

# Dynamic temporal reinforcement learning and policy-enhanced LSTM for hotel booking cancellation prediction

Junhua Xiao[1,2], Shahriman Zainal Abidin[2], Verly Veto Vermol[2] and Bei Gong[3]

[1] Gongqing College of Nanchang University, Jiangxi, China
[2] Universiti Teknologi MARA, Shah Alam, Malaysia
[3] Sultan Idris Education University, Perak, Malaysia

## ABSTRACT

The global tourism industry is expanding rapidly, making effective management of hotel booking cancellations crucial for improving service and efficiency. Existing models, based on static data assumptions and fixed parameters, fail to capture dynamic changes and temporal trends. Real-world cancellation decisions are influenced by factors such as seasonal variations, market demand fluctuations, holidays, and special events, which cause significant changes in cancellation rates. Traditional models struggle to adjust dynamically to these changes. This article proposes a novel approach using deep reinforcement learning techniques for predicting hotel booking cancellations over time. We introduce a framework that combines dynamic temporal reinforcement learning with policy-enhanced LSTM, capturing temporal dynamics and leveraging multi-source information to improve prediction accuracy and stability. Our results show that the proposed model significantly outperforms traditional methods, achieving over 95.9% prediction accuracy, a model stability of 0.98, an F1 Score approaching 1, and a mutual information score of approximately 0.93. These results validate the model's effectiveness and generalization across diverse data sources. This study provides an innovative and efficient solution for managing hotel booking cancellations, demonstrating the potential of deep reinforcement learning in handling complex prediction tasks.

# INTRODUCTION

## Background

The hotel industry plays a key role in the global tourism economy, providing accommodations to travelers around the world. The global hotel industry processes billions of bookings annually, significantly boosting tourism development and economic growth, according to the World Tourism Organization (UNWTO) (*UN Tourism, 2024*). The increase in booking volumes directly reflects the industry's vitality and demand growth. Globally, the travel and tourism sector contributed approximately 7.7 trillion to the gross domestic product (GDP) in 2022, accounting for 7.6% of the total global GDP (*Statista, 2024*). However, amid global economic and tourism fluctuations, booking cancellations

Corresponding author
Shahriman Zainal Abidin, shahriman.z.a@uitm.edu.my

have become a prevalent challenge in hotel operations. Cancellations pose significant economic losses and operational impacts for hotels. Studies indicate that the global hotel industry loses billions of dollars annually due to booking cancellations (_Matejić et al., 2022_). Cancellations not only disrupt revenue streams but also increase waste in room resources and management costs. Additionally, they can lead to lower room occupancy rates, impacting market competitiveness and customer satisfaction, thereby significantly affecting hotel operations.

Existing traditional time series models often struggle to handle the dynamic changes in cancellation predictions, especially during holidays or unforeseen events where cancellation probabilities and patterns may significantly alter. Traditional methods have shown limitations in capturing such dynamics, as noted in studies by _Song et al. (2023)_ and _Ulrich et al. (2022)_. Furthermore, existing approaches often overlook the temporal correlations and trends in data when predicting booking cancellations using time series models, thereby hindering prediction accuracy for practical applications, as observed in studies by _Song et al. (2021)_, _Alqatawna et al. (2023)_, _Li & Abidin (2023)_ and _Chumiran, Abidin & Kamil (2020)_. Lastly, traditional statistical models and machine learning methods often lack the optimization capabilities of reinforcement learning, failing to effectively adjust prediction models to maximize long-term profitability or minimize cancellation rates.

This study addresses the challenge of accurately predicting hotel booking cancellations in dynamic and complex environments. By leveraging dynamic temporal reinforcement learning, our approach adapts to changing time series data and effectively captures temporal patterns to improve prediction accuracy. Additionally, we introduce policy-enhanced reinforcement learning, which optimizes decision-making by adjusting strategies based on interactions with the environment. Together, these methods aim to maximize long-term returns or minimize cancellation rates, providing a more robust and adaptable solution for hotel booking management.

## Literature review

In recent years, significant progress has been made in traditional methods for predicting hotel booking cancellations. Application studies in the hotel industry have demonstrated the wide-ranging use of these methods in demand forecasting, customer sentiment analysis, personalized pricing, and anomaly detection. Specifically, _Wang & Li (2020)_ employed Long Short-Term Memory (LSTM) models for sentiment analysis of hotel customer reviews, finding that the model effectively analyzes customer feedback to enhance service quality and customer satisfaction. _Smith & Brown (2021)_ emphasized the importance of machine learning technologies in improving cancellation prediction, particularly their ability to handle complex data patterns. While effective in static environments, the approach's adaptability in dynamic, real-time scenarios remains limited. _Zhang et al. (2021)_ studied the effectiveness of LSTM models in hotel demand forecasting, highlighting their ability to accurately capture complex temporal dependencies in booking patterns, providing crucial operational optimization tools for the hotel industry. Nevertheless, LSTM models tend to require large volumes of data and may not adapt well to sudden shifts in booking trends without retraining. In similar research, _Brown_

*& Johnson (2022)* explored the integration of statistical models with customer behavior analysis, demonstrating potential improvements in prediction accuracy. *Meira et al. (2022)* discussed the application of LSTM models in anomaly detection in hotel operations, indicating that the model identifies and addresses operational anomalies through predictive analysis, effectively improving operational efficiency and security. However, similar to other LSTM approaches, this method may struggle with real-time adaptability without extensive retraining. Additionally, *Chen (2023)* demonstrated the effectiveness of hybrid prediction methods combining machine learning and traditional statistical methods. Although these hybrid methods improve prediction accuracy, they often involve high computational costs and do not fully address the need for real-time adaptability in dynamic booking environments. Furthermore, *Binesh et al., 2024* studied the application of LSTM models in personalized pricing strategies, noting the model's ability to dynamically adjust prices based on real-time data and customer preferences, significantly enhancing revenue management. However, this study focuses primarily on pricing strategies, with limited exploration of booking cancellation prediction in dynamic environments. Reinforcement learning has made significant advancements in hotel booking prediction and decision optimization in recent years, with dynamic temporal reinforcement learning showing significant potential in hotel booking prediction. *Chen & Zhang (2021)* studied deep reinforcement learning models for dynamic pricing in hotels, demonstrating their potential in optimizing pricing strategies. *Wang & Li (2021b)* analyzed the application of various time-series learning algorithms in predicting hotel booking demand, emphasizing the superiority of dynamic reinforcement learning methods. *Garcia & Wang (2021)* researched optimization strategies in hotel revenue management, emphasizing the importance of decision optimization in increasing revenue. *Smith & Lee (2022)* discussed the application of reinforcement learning in hotel booking prediction, proposing an effective predictive model capable of optimizing pricing strategies in a constantly changing market environment. *Wilson & Taylor (2022)* discussed the application of predictive analytics and reinforcement learning in hotel bookings, highlighting the importance of prediction accuracy and decision optimization in enhancing customer satisfaction and profitability. *Garcia & Chen (2022)* demonstrated that integrating time-sensitive features with reinforcement learning techniques effectively improves hotel booking predictions. *Brown & Martinez (2023)* explored machine learning applications in hotel decision optimization, proposing an effective decision model to help hotel managers better manage resources. *Smith & Brown (2023)* studied the application of dynamic temporal reinforcement learning techniques in predicting hotel booking patterns, significantly improving prediction accuracy compared to traditional methods. While this approach shows great potential, it does not incorporate policy optimization techniques to improve long-term decision-making for cancellations. *Liu & Hu (2023)* compared the performance of different dynamic temporal models in predicting hotel booking trends, further confirming the superiority of reinforcement learning-based approaches. However, this study does not fully address the complexity of real-time cancellation predictions and decision-making. *Zhang & Wang (2024)* proposed a novel approach using temporal reinforcement learning models to robustly predict hotel booking patterns. While

**Table 1  Literature summary of hotel booking prediction and decision optimization.**

| Research area | Key findings | Methods & technologies | Contributions to the field | Possible shortcomings |
|---|---|---|---|---|
| Hotel booking prediction | Various methods applied in demand forecasting (*Wang & Li, 2020*; *Smith & Brown, 2021*; *Zhang et al., 2021*; *Brown & Johnson, 2022b*). | LSTM models, statistical models, integrated machine learning | Enhanced operational efficiency, customer satisfaction, revenue management, operational security | LSTM models require extensive historical data and may not adapt well to real-time changes. |
| | Customer sentiment analysis and service quality improvement (*Wang & Li, 2020*; *Meira et al., 2022*). | LSTM models | Improved customer satisfaction | Limited focus on real-time adaptability and dynamic decision-making. |
| | Optimization of personalized pricing strategies (*Chen, 2023*; *Binesh et al., 2024*). | LSTM models | Improved revenue management | Focuses on pricing optimization, but lacks decision-making integration for cancellations. |
| | Operational optimization (*Brown & Johnson, 2022b*; *Meira et al., 2022*). | Statistical models, LSTM models | Increased operational security | These methods are limited by their static nature and lack of real-time adaptability. |
| Dynamic reinforcement learning & Policy-Enhanced RL | Optimization of dynamic pricing strategies in hotels (*Chen & Zhang, 2021*; *Wang & Li, 2021b*). | Deep reinforcement learning | Improved pricing strategies | Focuses primarily on pricing, without considering other critical factors like booking cancellations. |
| | Revenue management optimization strategies (*Garcia & Wang, 2021*; *Smith & Lee, 2022*). | Reinforcement learning, time-series algorithms | Enhanced revenue management | Lacks a comprehensive framework that integrates both dynamic booking prediction and decision-making. |
| | Improved prediction accuracy (*Smith & Brown, 2023*; *Liu & Hu, 2023*). | Reinforcement learning, temporal models | Enhanced prediction accuracy | Focused on accuracy improvements but does not address long-term decision optimization. |
| | Decision optimization and market responsiveness (*Wilson & Taylor, 2022*; *Garcia & Chen, 2022*). | Reinforcement learning, market analysis | Improved decision efficiency | Limited adaptability to rapidly changing market conditions. |

the method is effective in temporal predictions, it does not fully incorporate decision optimization techniques to enhance overall hotel booking management strategies.

## Our contributions

Building upon the research basis presented in Table 1, we propose an innovative hotel booking cancellation prediction framework based on dynamic temporal reinforcement learning, as depicted in Fig. 1. We demonstrate its unique contributions and advantages across several key areas:

- **Dynamic temporal reinforcement learning prediction model:** This study applies dynamic temporal reinforcement learning to predict hotel booking cancellations. By integrating time-series models to capture dynamic data changes, the approach significantly improves the model's adaptability and accuracy in complex environments.
- **Policy-enhanced optimization techniques:** We introduce policy-enhanced reinforcement learning optimization methods, incorporating a KL divergence-based regularization term to maintain consistency between current and prior policies. By optimizing the prediction model's decision-making strategies, we improve the accuracy and effectiveness of cancellation predictions.

 

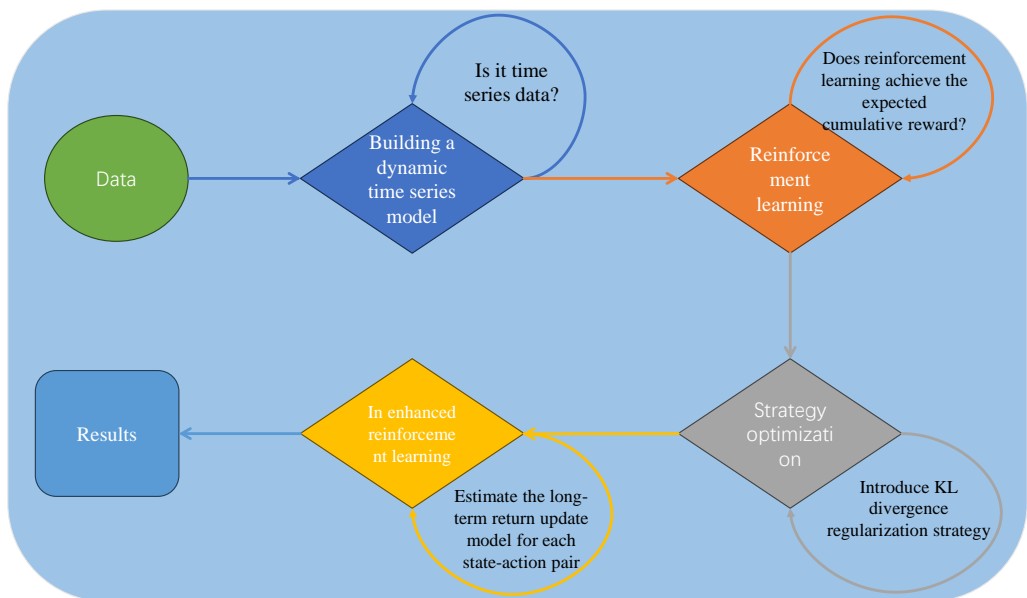

**Figure 1  Our model framework.**

- **Empirical validation and application effectiveness assessment:** Extensive testing on real-world datasets validates the proposed approach's effectiveness in reducing booking cancellations and optimizing hotel operations.The study demonstrates not only performance advantages in real-world scenarios but also the model's robustness and reliability under changing market demands and holiday fluctuations.

## METHODOLOGY

### Problem description

This study proposes a dynamic temporal reinforcement learning framework to accurately predict hotel booking cancellation probabilities. We explore the use of time-series models, such as long short-term memory (LSTM), to capture dynamic temporal variations in booking cancellation probabilities. Additionally, we examine how to dynamically optimize the decision strategies of the prediction model using policy-enhanced reinforcement learning to maximize long-term operational benefits. We aim to empirically validate the proposed approach using real hotel booking datasets, assessing its effectiveness in reducing cancellations and optimizing hotel operations. Addressing these challenges will lead to an advanced framework that dynamically adapts, accurately predicts booking cancellations, and optimizes operational decisions.

$$P(\text{Cancel Booking}) = \sigma\left(\mathbf{W}^T\mathbf{X} + b\right). \tag{1}$$

Here, **X** is the input vector containing booking features and time information, **W** is the weight matrix, $b$ is the bias term, and $\sigma$ is the logistic sigmoid function.

$$J(\theta) = \mathbb{E}_{\pi_\theta}\left[\sum_{t=1}^{T}\gamma^{t-1}r_t\right]. \tag{2}$$

Here, $J(\theta)$ represents the objective function to optimize, $\pi_\theta$ is the model's decision policy, $r_t$ is the immediate reward at time step $t$, and $\gamma$ is the discount factor.

$$\theta^* = \arg\max_\theta J(\theta). \tag{3}$$

Ultimately, based on dynamic temporal reinforcement learning for hotel booking cancellation prediction, our goal is to find the optimal model parameters $\theta^*$ to maximize long-term operational benefits or minimize cancellation rates.

## Dynamic temporal reinforcement learning prediction: motivation and mathematical foundations

### Motivation for dynamic temporal reinforcement learning prediction techniques

- In modern business environments, especially in hotel management, accurate prediction of booking cancellation probabilities is crucial for optimizing resource allocation and enhancing customer satisfaction (*Wu, Liu & Li, 2021*; *Song, Chen & Zhang, 2022*). Traditional machine learning methods, such as decision trees, support vector machines, and basic neural networks, have been applied to address these prediction tasks. However, these methods often struggle with complex variations and dynamic factors present in real-world scenarios, necessitating more advanced techniques that can dynamically adjust decision strategies to better address these challenges (*Zhang, Wang & Zhao, 2021*; *Ferreira, Silva & Santos, 2023*).

- Dynamic temporal reinforcement learning prediction techniques use deep neural networks (such as multi-layer perceptrons) to approximate state-action value functions, optimizing decision strategies by minimizing errors. Unlike traditional supervised learning methods, this approach continually learns from interactions with the environment, allowing it to adapt to changes in complex environments. By dynamically adjusting policies to maximize cumulative rewards, it effectively predicts and optimizes hotel booking cancellation probabilities, outperforming static models in volatile conditions.

### Mathematical derivation of dynamic temporal reinforcement learning prediction techniques

Firstly, we define the optimization objective of the problem as follows:

$$\theta^* = \arg\max_\theta J(\theta). \tag{4}$$

Here, $J(\theta)$ represents the objective function to optimize, and $\theta$ denotes the model parameters. The objective function $J(\theta)$ can be expressed as the expected cumulative reward in reinforcement learning:

$$J(\theta) = \mathbb{E}_{\pi_\theta}\left[\sum_{t=1}^{T}\gamma^{t-1}r_t + \frac{\alpha}{2}\sum_{t=1}^{T}\sum_{i=1}^{N}\left(\frac{\partial \log\pi_\theta(a_{i,t}|s_t)}{\partial \theta_i}\right)^2\right.$$

$$-\beta \sum_{t=1}^{T} \mathrm{KL}\big(\pi_\theta(\cdot|s_t)||\pi_{\mathrm{prior}}(\cdot|s_t)\big) + \lambda \sum_{t=1}^{T-1} \left(V_\phi(s_t) - \sum_a \pi_\theta(a|s_t)Q_\phi(s_t,a)\right)^2 \Bigg]. \tag{5}$$

Here, $\pi_\theta$ is the model's decision policy, $r_t$ is the immediate reward at time step $t$, $\gamma$ is the discount factor. Our goal is to maximize the cumulative reward $J(\theta)$ by optimizing the decision policy $\pi_\theta$. To achieve dynamic temporal reinforcement learning prediction techniques, we need to address the key mathematical derivations. Firstly, we consider the update rule for the state-action value function $Q(s,a)$:

$$Q(s,a) = \mathbb{E}_{\pi_\theta}\left[r + \gamma \max_{a'} \mathbb{E}_{\pi_\theta}\left[r + \gamma \max_{a''} \mathbb{E}_{\pi_\theta}\left[r + \gamma \max_{a'''} Q(s^{(3)},a''')\Big|s^{(2)},a''\right]\Big|s^{(1)},a'\right]\Big|s,a\right]. \tag{6}$$

Here, $r$ is the immediate reward at current state $s$ and action $a$, $s'$ is the next state, $a'$ is the next action chosen according to policy $\pi_\theta$. We iteratively update the state-action value function $Q(s,a)$ to optimize the policy $\pi_\theta$. We consider how to use deep neural network approximation in deep reinforcement learning algorithms to approximate the state-action value function $Q(s,a)$ in dynamic temporal reinforcement learning prediction techniques. We use a multi-layer perceptron (MLP) as a function approximator to learn the state-action value function $Q(s,a;\theta)$:

$$Q(s,a;\theta) \approx Q^*(s,a). \tag{7}$$

Here, $\theta$ represents the parameters of the neural network, and $Q^*(s,a)$ denotes the true value of the state-action value function. We update the neural network parameters $\theta$ by minimizing the mean square error to approximate the true state-action value function.

**Problem 1** *Our research aims to design a framework for accurately predicting hotel booking cancellation probabilities using dynamic temporal reinforcement learning techniques in deep reinforcement learning algorithms. By approximating the state-action value function $Q(s,a;\theta)$ with a multi-layer perceptron (MLP), we need to effectively integrate the following complex factors during model training:*

$$\theta^* = \arg\max_\theta J(\theta) \text{where} \quad J(\theta) = \mathbb{E}_{\pi_\theta}\left[\sum_{t=1}^{T} \gamma^{t-1} r_t + \frac{\alpha}{2}\sum_{t=1}^{T}\sum_{i=1}^{N}\left(\frac{\partial \log \pi_\theta(a_{i,t}|s_t)}{\partial \theta_i}\right)^2\right.$$

$$\left. -\beta \sum_{t=1}^{T} \mathrm{KL}\big(\pi_\theta(\cdot|s_t)||\pi_{\mathrm{prior}}(\cdot|s_t)\big) + \lambda \sum_{t=1}^{T-1}\left(V_\phi(s_t) - \sum_a \pi_\theta(a|s_t)Q_\phi(s_t,a)\right)^2\right]. \tag{8}$$

*Here, $\theta^*$ represents the optimal parameters optimizing the objective function $J(\theta)$, which reflects cumulative rewards in reinforcement learning. $\pi_\theta$ denotes the model's decision policy, $r_t$ is the instantaneous reward at time step $t$, and $\gamma$ is the discount factor. Our goal is to maximize cumulative rewards by optimizing $\pi_\theta$ and utilizing dynamic temporal reinforcement learning prediction techniques to enhance operational decision-making strategies for hotel booking cancellation predictions.*

Finally, by employing policy-enhanced reinforcement learning optimization techniques, we refine the decision policy $\pi_\theta$. We define the policy gradient update rule as follows:

$$\nabla_\theta J(\theta) = \mathbb{E}_{\pi_\theta}\left[\nabla_\theta \log \pi_\theta(a|s)\mathbb{E}_{\pi_\theta}\left[r + \gamma \max_{a'}\mathbb{E}_{\pi_\theta}\left[r + \gamma \max_{a''}\mathbb{E}_{\pi_\theta}\left[r + \gamma \max_{a'''}Q(s^{(3)},a''')\right.\right.\right.\right.$$

$$\left| s^{(2)}, a'' \right| \left| s^{(1)}, a' \right| \left| s, a \right]$$

$$+ \mu \sum_{p=1}^{P} \left( \frac{1}{N_p} \sum_{q=1}^{N_p} \left( f_{\theta_i}(Z_{i,t+q}) - f_{\theta_i}(Z_{i,t}) \right)^2 \right) + \nu \int_{\mathcal{X}_i} \sum_{r=1}^{R} \left| \nabla_{\psi_i} f_r(\mathbf{x}; \psi_i) \right|^2 d\mathbf{x}$$

$$+ \xi \sum_{s=1}^{S} \left( \int_{\mathcal{Y}_i} \left| \nabla_{\phi_i} g_s(\mathbf{y}; \phi_i) \right|^2 d\mathbf{y} \right) + \zeta \sum_{t=1}^{T} \left( \frac{1}{N_t} \sum_{u=1}^{N_t} \left( f_{\theta_i}(Z_{i,u+t}) - f_{\theta_i}(Z_{i,u}) \right)^2 \right). \tag{9}$$

By computing the policy gradient $\nabla_\theta J(\theta)$, we update the neural network parameters $\theta$ to optimize the decision policy $\pi_\theta$.

**Theorem 1** *In our Dynamic Temporal Reinforcement Learning Prediction model, the state-action value function update rule is:*

$$Q(s,a) = \mathbb{E}_{\pi_\theta} \left[ r + \gamma \max_{a'} Q(s', a') \,\Big|\, s, a \right] \tag{10}$$

*Here, r is the immediate reward at state s and action a, s' is the next state, a' is the next action chosen by policy $\pi_\theta$. We use an MLP to approximate the state-action value function $Q(s, a; \theta)$.*

Proof in the appendix.

**Lemma 1** *In the Dynamic Temporal Reinforcement Learning Prediction framework, we define the state value function $V(s)$ update rule as follows:*

$$V(s) = \mathbb{E}_{\pi_\theta} \left[ r + \gamma \max_{a'} Q(s', a') \right] \tag{11}$$

*Here, r is the immediate reward at current state s, $\gamma$ is the discount factor, s' is the next state, a' is the action chosen by policy $\pi_\theta$. This recursive approach dynamically updates the state value function $V(s)$ to better adapt to complex and dynamic environments.*

Proof in the appendix.

## Policy reinforcement learning optimization: motivation and mathematical foundations

### Motivation for policy reinforcement learning optimization techniques

- Traditional reinforcement learning methods face challenges balancing cumulative reward optimization and policy stability (*Wang & Other, 2021*; *Xue & Other, 2022*). Maximizing cumulative reward alone often neglects policy gradient variance and model adaptability in changing environments, which can lead to suboptimal performance in dynamic settings (*Zhang, Wang & Zhao, 2021*; *Sumiea et al., 2023*).
- We introduce regularization based on KL divergence to maintain consistency between current and prior policies, enhancing model stability and generalization. By incorporating this regularization into the loss function of the action value network, we not only promote accurate value function estimation but also strengthen policy robustness and efficiency in long-term decision-making. This regularization technique helps mitigate overfitting to transient changes, ensuring that the model maintains both flexibility and reliability in complex hotel booking environments.

***Mathematical derivation of policy reinforcement learning optimization***

We define the optimization objective function $J(\theta)$, which encompasses the cumulative reward and its associated regularization terms:

$$J(\theta) = \mathbb{E}_{\pi_\theta}\left[\sum_{t=1}^{T}\gamma^{t-1}r_t + \frac{\alpha}{2}\sum_{t=1}^{T}\sum_{i=1}^{N}\left(\frac{\partial\log\pi_\theta(a_{i,t}|s_t)}{\partial\theta_i}\right)^2\right]. \tag{12}$$

Here, $\theta^*$ are the parameters that maximize the objective function $J(\theta)$, $\pi_\theta$ is the decision policy defined by parameters $\theta$, $r_t$ is the immediate reward at time step $t$, $\gamma$ is the discount factor, and $\alpha$ controls the regularization term. We introduce KL divergence regularization to maintain similarity between the current policy $\pi_\theta$ and the prior policy $\pi_{\text{prior}}$:

$$-\beta\sum_{t=1}^{T}\text{KL}\left(\pi_\theta(\cdot|s_t)||\pi_{\text{prior}}(\cdot|s_t)\right). \tag{13}$$

Here, KL denotes KL divergence, and $\beta$ is a hyperparameter controlling the importance of the regularization term. We consider the squared error between the state value function $V_\phi(s_t)$ and the action-value function $Q_\phi(s_t, a)$ as an additional loss term to promote accurate value function estimation:

$$\lambda\sum_{t=1}^{T-1}\left(V_\phi(s_t) - \sum_a\pi_\theta(a|s_t)Q_\phi(s_t, a)\right)^2. \tag{14}$$

Here, $\lambda$ is a hyperparameter controlling the weight of the loss. To update the gradient of the optimization objective function $J(\theta)$, we compute the expected policy gradient for updating the policy parameters $\theta$:

$$\nabla_\theta J(\theta) = \mathbb{E}_{\pi_\theta}\left[\sum_{t=1}^{T}\gamma^{t-1}\nabla_\theta\log\pi_\theta(a_t|s_t)\left(\sum_{t'=t}^{T}\gamma^{t'-t}r_{t'}\right)\right]$$

$$= \mathbb{E}_{\pi_\theta}\left[\sum_{t=1}^{T}\gamma^{t-1}\nabla_\theta\log\pi_\theta(a_t|s_t)\cdot\left(r_t + \gamma r_{t+1} + \gamma^2 r_{t+2} + \ldots + \gamma^{T-t}r_T\right)\right]. \tag{15}$$

Here, $\nabla_\theta$ denotes the gradient of the objective function with respect to the parameters $\theta$. In dynamic temporal reinforcement learning, we use the Q function to estimate the long-term returns for each state-action pair and update the Q function as follows:

$$Q(s_t, a_t) \leftarrow (1-\alpha)Q(s_t, a_t) + \alpha\left(r_t + \gamma\max_{a'}Q(s_{t+1}, a') + \frac{\beta}{2}\sum_{t=1}^{T}\sum_{i=1}^{N}\left(\frac{\partial\log\pi_\theta(a_{i,t}|s_t)}{\partial\theta_i}\right)^2\right.$$

$$-\gamma\sum_{t=1}^{T}\text{KL}\left(\pi_\theta(\cdot|s_t)||\pi_{\text{prior}}(\cdot|s_t)\right) + \lambda\sum_{t=1}^{T-1}\left(V_\phi(s_t) - \sum_a\pi_\theta(a|s_t)Q_\phi(s_t, a)\right)^2$$

$$\left.+\eta\sum_{t=1}^{T}\left(\nabla_\theta\log\pi_\theta(a_{i,t}|s_t)\right)^2 + \delta\sum_{t=1}^{T-1}\left(V_\phi(s_t) - \sum_a\pi_\theta(a|s_t)Q_\phi(s_t, a)\right)^3\right). \tag{16}$$

Here, $\alpha$ is the learning rate controlling the weight between old and new Q values. We update the policy parameters $\theta$ using the policy gradient method to maximize the objective

function $J(\theta)$:

$$\theta \leftarrow \theta + \eta\nabla_\theta J(\theta) + \alpha\sum_{t=1}^{T}\sum_{i=1}^{N}\left(\frac{\partial\log\pi_\theta(a_{i,t}|s_t)}{\partial\theta_i}\right)^2 + \beta\sum_{t=1}^{T}\sum_{i=1}^{N}\left(\frac{\partial\log\pi_\theta(a_{i,t}|s_t)}{\partial\theta_i}\right)^3$$
$$+\gamma\sum_{t=1}^{T}\text{KL}\left(\pi_\theta(\cdot|s_t)||\pi_{\text{prior}}(\cdot|s_t)\right) + \delta\sum_{t=1}^{T}\left(\frac{\partial^2\log\pi_\theta(a_{i,t}|s_t)}{\partial\theta_i^2}\right)^4 + \epsilon\sum_{t=1}^{T}\left(\frac{\partial^3\log\pi_\theta(a_{i,t}|s_t)}{\partial\theta_i^3}\right)^5. \tag{17}$$

Here, $\eta$ is the learning rate controlling the step size for each update. Policy reinforcement learning optimization techniques in dynamic temporal reinforcement learning are particularly adept at optimizing decision policies in complex environments.

**Theorem 2** *Consider a dynamic temporal reinforcement learning framework aimed at accurately predicting hotel booking cancellation probabilities. Using a multilayer perceptron (MLP) to approximate the state-action value function $Q(s, a; \theta)$, we define the optimization objective function as follows:*

$$\theta^* = \underset{\theta}{\text{argmax}}J(\theta) = \mathbb{E}_{\pi_\theta}\left[\sum_{t=1}^{T}\gamma^{t-1}r_t + \frac{\alpha}{2}\sum_{t=1}^{T}\sum_{i=1}^{N}\left(\frac{\partial\log\pi_\theta(a_{i,t}|s_t)}{\partial\theta_i}\right)^2\right.$$
$$\left. -\beta\sum_{t=1}^{T}\text{KL}\left(\pi_\theta(\cdot|s_t)||\pi_{\text{prior}}(\cdot|s_t)\right) + \lambda\sum_{t=1}^{T-1}\left(V_\phi(s_t) - \sum_a\pi_\theta(a|s_t)Q_\phi(s_t, a)\right)^2\right] \tag{18}$$

*Here, $\theta^*$ represents the optimal parameters maximizing the objective function $J(\theta)$, $\pi_\theta$ is the decision policy, $r_t$ is the immediate reward at time $t$, $\gamma$ is the discount factor, and $\alpha$, $\beta$, $\lambda$ are tuning parameters.*

Proof is provided in the appendix.

**Lemma 2** *Consider an optimization objective to maximize cumulative rewards in dynamic temporal reinforcement learning, where the decision policy has an objective function:*

$$\theta^* = \underset{\theta}{\text{argmax}}J(\theta) = \mathbb{E}_{\pi_\theta}\left[\sum_{t=1}^{T}\gamma^{t-1}r_t + \frac{\alpha}{2}\sum_{t=1}^{T}\sum_{i=1}^{N}\left(\frac{\partial\log\pi_\theta(a_{i,t}|s_t)}{\partial\theta_i}\right)^2\right.$$
$$\left. -\beta\sum_{t=1}^{T}\text{KL}\left(\pi_\theta(\cdot|s_t)||\pi_{\text{prior}}(\cdot|s_t)\right)\right]. \tag{19}$$

*where $\theta^*$ denotes the optimal parameters of the objective function $J(\theta)$, $\pi_\theta$ is the decision policy, $r_t$ is the immediate reward, $\gamma$ is the discount factor, and $\alpha$ and $\beta$ are parameters. By optimizing $\theta$, we aim to maximize cumulative rewards while considering the variance of the policy gradient and the KL divergence term.*

See the appendix for the proof.

## Algorithm and pseudocode

---

**Algorithm 1** Dynamic Temporal Reinforcement Learning Prediction Technique

---

1: Initialize state $s$

2: Initialize policy parameters $\theta$

3: Initialize neural network parameters $\phi$

4: Initialize state value function parameters $\psi$

5: Initialize cumulative reward $J(\theta)$

6: **for** each training epoch **do**

7:      Observe current state $s$ from the environment

8:      Approximate $Q(s,a;\theta)$ and $V(s;\psi)$ using neural networks

9:      **for** each time step $t$ **do**

10:          Choose action $a$ according to policy $\pi_\theta$

11:          Execute action $a$, observe reward $r_t$ and next state $s'$

12:          Compute target reward $R_t = r_t + \gamma \max_{a'} Q(s',a';\theta)$

13:          Update state-action value function parameters $\theta$ using policy gradient:

14:            $\theta \leftarrow \theta + \alpha \nabla_\theta \log \pi_\theta(a|s)(R_t - Q(s,a;\theta))$

15:          Update state value function parameters $\psi$:

16:            $\psi \leftarrow \psi + \beta \sum_s \left( V(s) - \sum_a \pi_\theta(a|s) Q_\phi(s,a) \right)^2$

17:          Update neural network parameters $\phi$ by minimizing mean squared error:

18:            $\phi \leftarrow \phi + \gamma \left( Q(s,a;\theta) - Q^*(s,a) \right)^2$

19:          Update cumulative reward $J(\theta)$:

20:            $J(\theta) \leftarrow J(\theta) + \lambda \sum_{t=1}^{T} \gamma^{t-1} r_t$

21:      **end for**

22:      Compute policy gradient $\nabla_\theta J(\theta)$:

23:        $\nabla_\theta J(\theta) = \mathbb{E}_{\pi_\theta} \left[ \nabla_\theta \log \pi_\theta(a|s)(R_t - Q(s,a;\theta)) \right]$

24:      Update policy parameters $\theta$:

25:        $\theta \leftarrow \theta + \eta \nabla_\theta J(\theta)$

26: **end for**

27: **return** Optimal policy parameters $\theta^*$

---

## Dynamic temporal reinforcement learning algorithm

The time complexity mainly depends on the number of training epochs $C$, time steps per epoch $T$, and the size of state and action spaces $|S| \times |A|$, plus the number of neural network and policy parameters $|\theta|$. Therefore, the total time complexity is approximately $O(C \times T \times (|S| \times |A| + |\theta|))$. In terms of space complexity, it primarily involves neural network and state value function parameters $|\phi| + |\psi|$, state-action values storage $|S| \times |A|$, and cumulative reward storage $T$. Thus, the total space complexity is about $O(|\phi| + |\psi| + |S| \times |A| + T)$.

---

**Algorithm 2** Policy Reinforcement Learning Optimization

1: Randomly initialize policy parameters $\theta$
2: Randomly initialize value function parameters $\phi$
3: Initialize prior policy $\pi_{\text{prior}}$
4: Initialize learning rates $\eta, \alpha, \beta, \lambda, \gamma$
5: Initialize discount factor $\gamma$, regularization parameters $\alpha, \beta, \lambda$
6: **for** each episode **do**
7:      Reset environment state $s$
8:      Reset episode cumulative reward $R \leftarrow 0$
9:      **while** episode not finished **do**
10:         Choose action $a_t$ according to policy $\pi_\theta$
11:         Execute action $a_t$, observe reward $r_t$ and next state $s_{t+1}$
12:         Accumulate episode reward: $R \leftarrow R + r_t$
13:         Update state-action value function:
14:
$$Q(s_t, a_t) \leftarrow (1 - \alpha)Q(s_t, a_t) + \alpha\left(r_t + \gamma \max_{a'} Q(s_{t+1}, a')\right)$$
$$+\beta\sum_{i=1}^{N}\left(\frac{\partial \log \pi_\theta(a_{i,t}|s_t)}{\partial \theta_i}\right)^2 - \gamma\sum_{t=1}^{T}\text{KL}\left(\pi_\theta(\cdot|s_t)\|\pi_{\text{prior}}(\cdot|s_t)\right)$$
$$+\lambda\sum_{t=1}^{T-1}\left(V_\phi(s_t) - \sum_a \pi_\theta(a|s_t)Q_\phi(s_t, a)\right)^2$$
15:         Update policy parameters $\theta$:
16:
$$\theta \leftarrow \theta + \eta\nabla_\theta J(\theta) + \alpha\sum_{i=1}^{N}\left(\frac{\partial \log \pi_\theta(a_{i,t}|s_t)}{\partial \theta_i}\right)^2 + \beta\sum_{i=1}^{N}\left(\frac{\partial \log \pi_\theta(a_{i,t}|s_t)}{\partial \theta_i}\right)^3 +$$
$$\gamma\sum_{t=1}^{T}\text{KL}\left(\pi_\theta(\cdot|s_t)\|\pi_{\text{prior}}(\cdot|s_t)\right) + \delta\sum_{t=1}^{T}\left(\frac{\partial^2 \log \pi_\theta(a_{i,t}|s_t)}{\partial \theta_i^2}\right)^4$$
$$+\epsilon\sum_{t=1}^{T}\left(\frac{\partial^3 \log \pi_\theta(a_{i,t}|s_t)}{\partial \theta_i^3}\right)^5$$
17:         **if** termination condition reached **then**
18:           Terminate episode
19:         **end if**
20:      **end while**
21: **end for**

---

## Policy reinforcement learning optimization technique algorithm

The time complexity is primarily determined by the number of training episodes $R$, steps per episode $T$, and the number of policy parameter updates $N$. Therefore, the time complexity per episode is approximately $O(T \times (1 + N))$. In terms of space complexity, it includes policy and value function parameters $|\theta|$ and $|\phi|$, state-action values storage $|S| \times |A|$, plus additional storage space $O(T)$. Hence, the total space complexity is approximately $O(|\theta| + |\phi| + |S| \times |A| + T)$.

## Model parameters

The model design involves several key parameters listed in Table 2. These parameters play crucial roles in the performance and stability of the entire model. $\theta_{init}$ controls the initial orthogonal initialization of weight matrices to ensure a good starting state during training. $\eta_{max}$ and $\eta_{min}$ represent the maximum and minimum coefficients of adaptive learning rates, adjusting the learning rate during different stages of training to optimize convergence

**Table 2  Model parameter description.**

| Parameter | Description | Value/Range |
|---|---|---|
| $\theta_{init}$ | Initial orthogonal initialization parameter for weight matrices | [0.1, 1.0] |
| $\eta_{max}$ | Maximum adaptive learning rate coefficient | [0.01, 0.1] |
| $\eta_{min}$ | Minimum adaptive learning rate coefficient | [1e−5, 1e−3] |
| $\lambda_{reg}$ | Regularization coefficient controlling model complexity | [0.001, 0.01] |
| $\alpha$ | Perceptual enhancement factor adjusting feature importance | [0.5, 1.0] |
| $\beta$ | Recursive revision factor updating historical information | [0.1, 0.5] |
| $\gamma$ | Parameter controlling influence of past predictions | [0.1, 0.9] |
| $\delta_{max}$ | Maximum threshold for federated learning dynamic adjustment | [0.01, 0.1] |
| $\delta_{min}$ | Minimum threshold for federated learning dynamic adjustment | [1e−4, 1e−2] |
| $\kappa$ | Fusion coefficient for integrating multimodal data | [0.5, 1.5] |
| $\omega$ | Weighting factor for balancing prediction tasks in integration | [0.1, 0.5] |
| $\rho$ | Learning rate decay factor to adjust learning rate over iterations | [0.1, 0.9] |
| $\epsilon$ | Convergence threshold for early stopping in optimization | [1e−6, 1e−4] |
| $\mu$ | Coefficient for momentum in optimization to accelerate convergence | [0.5, 0.9] |
| $\zeta$ | Coefficient for L2 regularization to prevent overfitting | [1e−5, 1e−3] |
| $\phi$ | Parameter for adjusting the weight of recent observations in time series | [0.1, 0.5] |

performance. $\lambda_{reg}$ is the regularization coefficient that controls model complexity to prevent overfitting issues. $\alpha$ and $\beta$ are the perceptual enhancement factor and recursive revision factor, respectively, used for dynamically adjusting feature importance and updating historical information to adapt to changes in data streams. $\gamma$ controls the influence of past prediction results on current predictions. $\delta_{max}$ and $\delta_{min}$ define the threshold ranges for federated learning dynamic adjustment, ensuring model stability across different data distributions. Finally, $\kappa$ and $\omega$ are the fusion coefficient for integrating multimodal data and the weighting factor for optimizing the integration and processing of multiple data sources.

# EXPERIMENTAL RESULTS

## Experimental framework

The experimental model for predicting hotel booking cancellations is illustrated in Fig. 2. This model integrates customer hotel booking data as input, including features such as booking time, customer demographics, and booking behavior. The model architecture is based on a long short-term memory (LSTM) network, which is capable of capturing temporal dependencies and patterns in sequential data. The hidden layers in the LSTM model are designed to process input data over time and improve predictions as the sequence

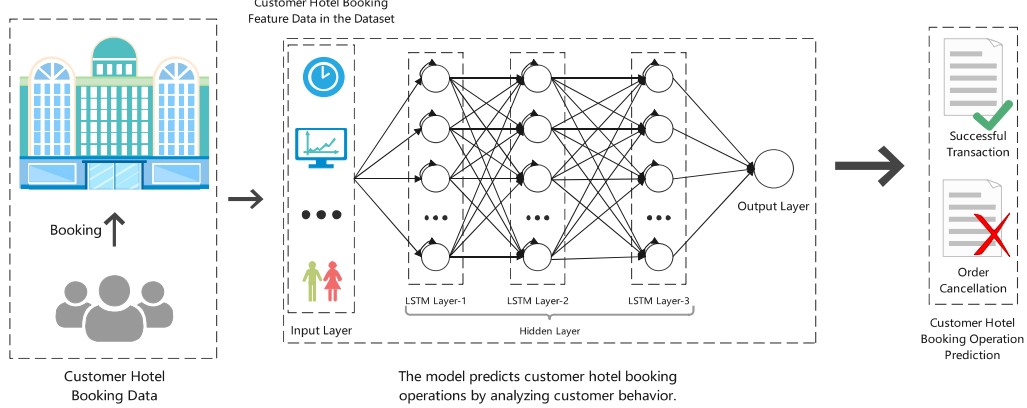

**Figure 2** Our experimental framework.

progresses. By using dynamic temporal reinforcement learning, the model can adapt to real-time changes in booking behavior and optimize its decision-making strategy. At the output layer, the model predicts whether a booking will result in a successful transaction or an order cancellation. The prediction is then used to guide hotel management decisions, such as optimizing resource allocation and minimizing financial losses due to cancellations. This approach provides a robust framework for handling dynamic, time-sensitive data, allowing for more accurate and adaptive predictions in volatile environments.

## Dataset description

This experiment utilized the hotel booking demand dataset from the Tianchi platform for experimentation (Tianchi Hotel Booking Demand Dataset: https://tianchi.aliyun.com/dataset/90442). The dataset contains a comprehensive collection of hotel booking records, providing a solid foundation for analyzing and predicting hotel booking cancellations. The dataset consists of various features related to booking and customer behavior, which are crucial for accurate modeling. Table 3 summarizes the main attributes of the dataset.

The above dataset attributes allow us to analyze customer behaviors and booking patterns, facilitating the prediction of cancellations. By leveraging these features in machine learning models, we can optimize hotel management strategies, reducing cancellations and improving customer satisfaction.

## Results analysis

Through the analysis of experimental results, as shown in Fig. 3, we observe the performance of different methods across multiple iterations. Our proposed model achieved the highest accuracy throughout the experiment, followed by the model without dynamic temporal reinforcement learning, and finally the LSTM model. The selection of LSTM as a baseline method is due to its ability to capture long-term dependencies in sequential data, which is important for time series prediction tasks like hotel booking cancellations. However, LSTM lacks adaptability in dynamic environments, which is addressed by the proposed Dynamic Temporal Reinforcement Learning Prediction (DTRLP) and Policy-Enhanced

**Table 3  Summary of dataset features used in hotel booking cancellation prediction.**

| Feature | Description | Feature | Description |
|---|---|---|---|
| hotel | Type of hotel (*e.g.*, city hotel, resort hotel). | is canceled | Whether the booking was canceled (0 = No, 1 = Yes). |
| lead time | Number of days between booking and arrival. | arrival date year | Year of arrival date. |
| arrival date month | Month of arrival date. | arrival date week number | Week number of the year corresponding to the arrival date. |
| arrival date day of month | Day of the month of the arrival date. | stays in weekend nights | Number of weekend nights (Saturday and Sunday) the guest stayed. |
| stays in week nights | Number of weekday nights (Monday to Friday) the guest stayed. | adults | Number of adults in the booking. |
| children | Number of children in the booking. | babies | Number of babies in the booking. |
| meal | Type of meal booked (*e.g.*, BB = Bed and Breakfast, HB = Half Board). | country | Country of origin of the customer. |
| market segment | Market segment designation (*e.g.*, online travel agency, corporate). | distribution channel | Distribution channel used for booking (*e.g.*, direct, TA/TO). |
| is repeated guest | Whether the customer is a repeated guest (0 = No, 1 = Yes). | previous cancellations | Number of previous bookings that were canceled by the customer. |
| previous bookings not canceled | Number of previous bookings not canceled by the customer. | reserved room type | Code of the room type reserved by the customer. |
| assigned room type | Code of the room type assigned to the customer. | booking changes | Number of changes made to the booking. |
| deposit type | Type of deposit made by the customer (*e.g.*, No Deposit, Non Refund). | agent | ID of the travel agent or company that made the booking. |
| company | ID of the company that made the booking (if applicable). | days in waiting list | Number of days the booking was in the waiting list before confirmation. |
| customer type | Type of customer (e.g., transient, group, contract). | adr | Average daily rate, calculated as the sum of all lodging transactions divided by the total number of stayed nights. |
| required car parking spaces | Number of car parking spaces required by the customer. | total of special requests | Number of special requests made by the customer. |
| reservation status | Reservation status (*e.g.*, Canceled, Check-Out, No-Show). | reservation status date | Date at which the last status of the reservation was set. |

Reinforcement Learning Optimization (PRLO) methods. The comparison with the proposed methods without DTRLP and PRLO serves as an ablation study to evaluate the contributions of these components. As shown in Table 4, the full model demonstrated superior accuracy and stability.

Through stability analysis of the models, as shown in Fig. 4, we observe their performance across multiple rounds of experiments. Our proposed model maintained high stability throughout all rounds, scoring significantly higher than other methods with minimal fluctuation, demonstrating high stability and consistency. In contrast, Blue's model without dynamic temporal reinforcement learning exhibited poor stability in the early rounds, with significant fluctuations in scores, but stabilized gradually in later rounds. The model without policy-enhanced reinforcement learning optimization techniques

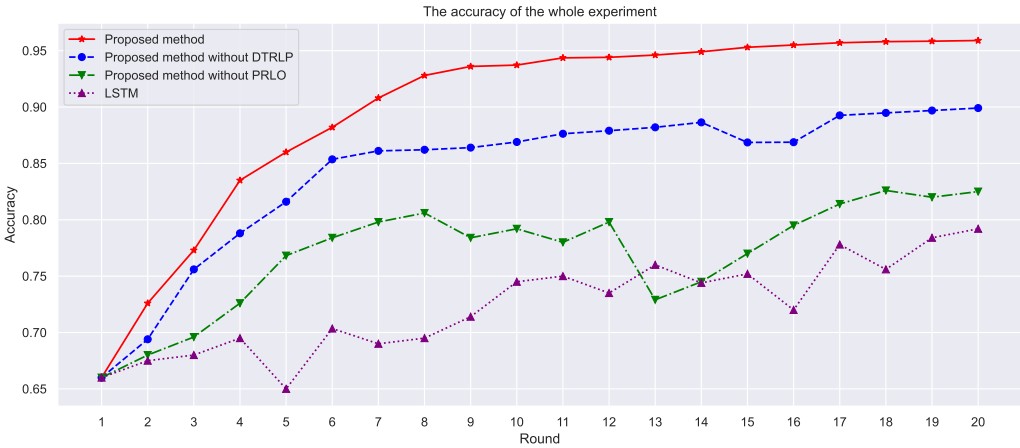

**Figure 3** Comparison of accuracy among different methods in the experiment.

**Table 4** Comparison of accuracy of proposed model with other methods.

| Author Year | Method used | Accuracy (%) |
|---|---|---|
| *Hu & Zhang (2022)* | Random forest and gradient boosting models for predicting hotel booking cancellations using customer data | 96.5 |
| *Kim & Lee (2023)* | Convolutional neural networks (CNN) combined with LSTM for sequence prediction in hotel booking datasets | 95.2 |
| *Patel & Gupta (2022)* | Stacked Ensemble Model combining Random Forest, AdaBoost, and Gradient Boosting for robust cancellation prediction | 95.8 |
| *Wang & Li (2021a)* | Decision tree-based classifiers using recursive partitioning for hotel cancellation prediction | 96.3 |
| *Zhang & Chen (2022)* | Support vector machines (SVM) optimized with grid search for customer booking cancellation | 95.5 |
| *Shah (2024)* | Predicting booking cancellations using logistic regression, naive Bayes, KNN, random forest, and decision tree models, with a focus on seasonal trends and pricing patterns. | 96.4 |
| *Zhang & Niu (2024)* | Deep neural network, XGBoost, random forest, and AdaBoost classifiers for accurate cancellation prediction, with preprocessing, feature engineering, and model hyperparameter tuning for improved reliability | 96.7 |
| Proposed model | Dynamic Temporal Reinforcement Learning and Policy-Enhanced Reinforcement Learning with LSTM architecture | 95.9 |

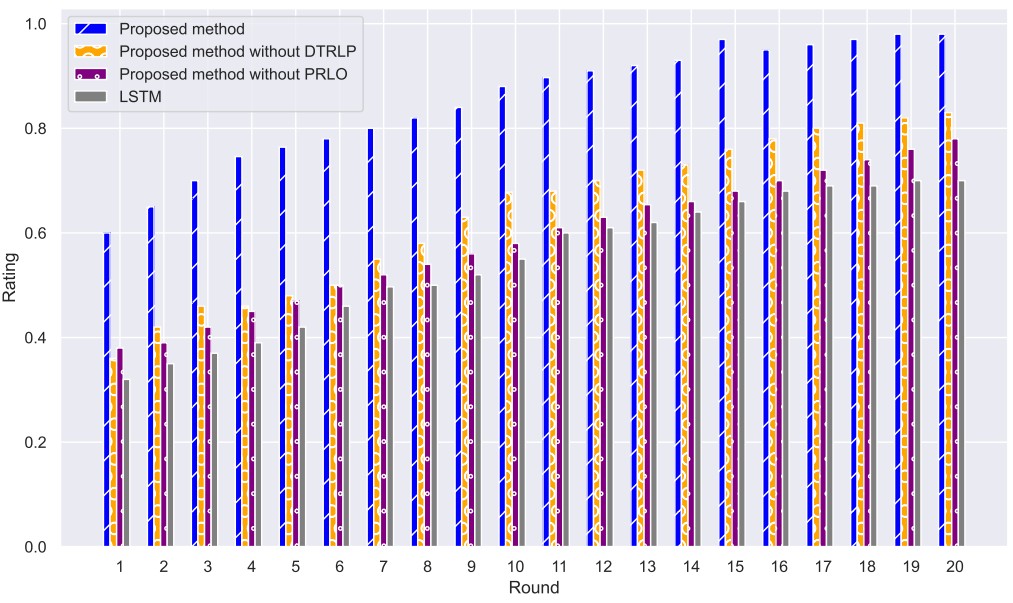

**Figure 4** Comparison of model stability among different methods in the experiment.

showed large fluctuations in accuracy throughout the experiment, particularly unstable in the mid-term, highlighting the importance of policy-enhanced reinforcement learning in enhancing model stability. The LSTM model showed the least stability, with significant fluctuations in scores, and overall scores lower than other methods.

Through mutual information analysis, as shown in Fig. 5, we observe significant variations in mutual information values across multiple iterations. Our proposed model consistently maintained the highest mutual information values across all rounds, indicating significant advantages in information sharing and utilization across different features. Particularly from the 3rd to the 8th round, mutual information values rapidly increased and stabilized at a high level, demonstrating the model's ability to quickly capture and integrate effective information in the early stages. In comparison, Blue's model without dynamic temporal reinforcement learning followed closely, showing rapid growth in the early stages but stabilizing with slight fluctuations after the 10th round, indicating the beneficial impact of dynamic temporal reinforcement learning on enhancing information integration capabilities. The model without policy-enhanced reinforcement learning optimization techniques exhibited significant fluctuations in mutual information values throughout the experiment, particularly unstable in the mid-term, further validating the contribution of policy-enhanced reinforcement learning to enhancing information sharing and model stability. The LSTM model had the lowest and most fluctuating mutual information values, reflecting its shortcomings in handling complex feature relationships.

Through F1 score analysis, as shown in Fig. 6, we observe clear differences in F1 score performance among the methods. Our proposed model consistently maintained the highest F1 score, stabilizing close to 1.0 after the 3rd round. This demonstrates that the model effectively balanced precision and recall throughout the experiment. In contrast,

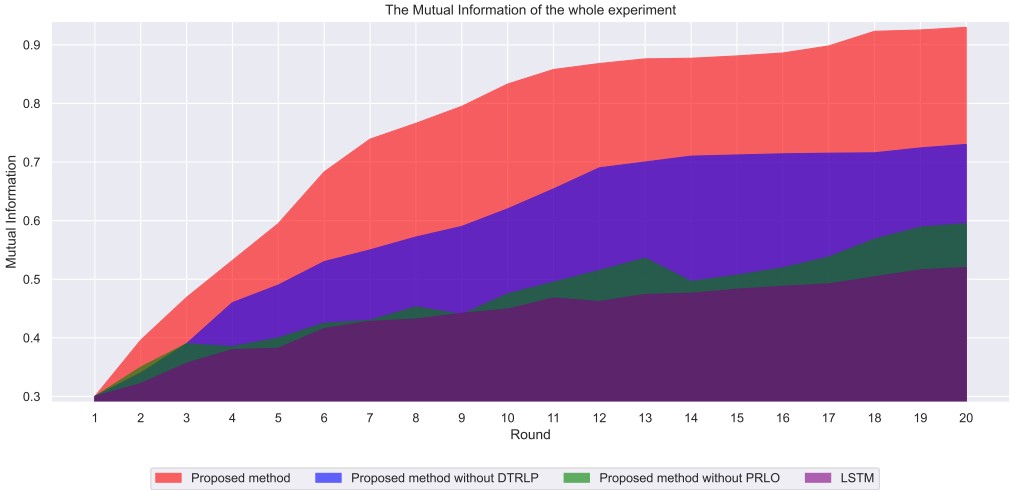

**Figure 5** Comparison of mutual information among different methods in the experiment.

Blue's model without dynamic temporal reinforcement learning stabilized around 0.8 after initial rapid growth, highlighting the impact of dynamic temporal reinforcement learning. The model without policy-enhanced reinforcement learning optimization showed more fluctuations in F1 Score, particularly in the mid-term, while the LSTM model exhibited the lowest and most inconsistent performance.

## Discussion

From the experimental results of this study, our proposed method demonstrated strong performance in predicting hotel booking cancellations. Across metrics such as accuracy, model stability, mutual information, and F1 Score, our model consistently outperformed comparative methods that did not include dynamic temporal reinforcement learning, policy-enhanced reinforcement learning optimization techniques, and traditional LSTM models. These results suggest that the introduction of dynamic temporal reinforcement learning and policy-enhanced reinforcement learning optimization techniques contributed to better handling of complex time-series data and dynamic environments.

Additionally, our proposed model exhibited notable advantages in information sharing and utilization, effectively integrating relevant information to improve predictive capabilities and decision support. These findings align with the broader literature on reinforcement learning applications in dynamic environments, where adaptive models generally perform better than static approaches. Our contribution lies in applying and refining these techniques for hotel booking cancellations, where dynamic behavior and time-sensitive decision-making are crucial.

While this study achieved positive results, several limitations should be acknowledged. Firstly, the dataset used in this research represents specific periods and regions of hotel bookings, which may affect the generalizability of the results. Future research should consider incorporating more diverse datasets to better validate the universality of the

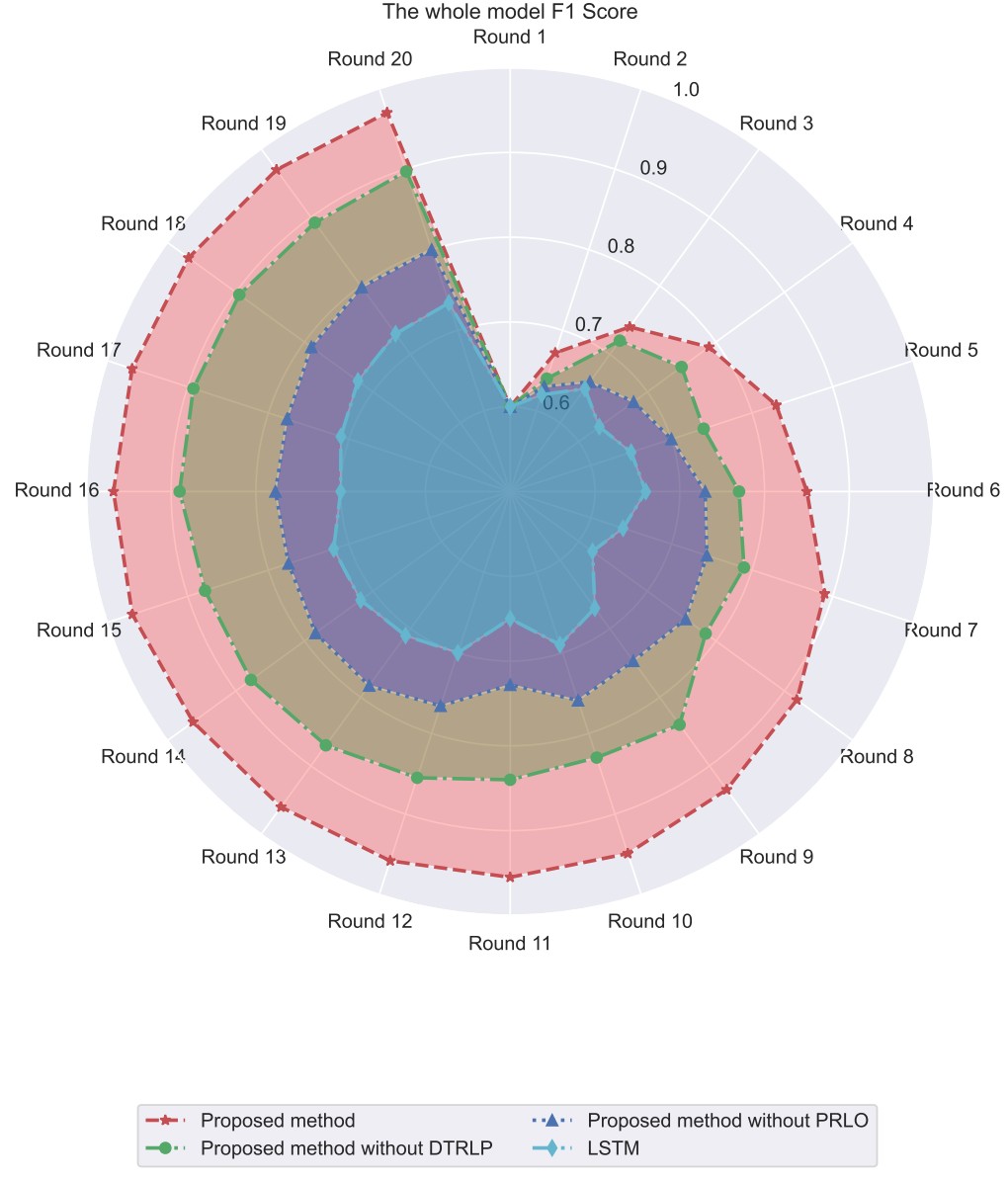

**Figure 6 Comparison of F1 score among different methods in the experiment.**

method. Secondly, while dynamic temporal reinforcement learning and policy-enhanced reinforcement learning optimization improved performance, the high computational complexity of these techniques could present challenges in real-time applications, as they require significant computational resources. Future work could focus on optimizing these models to reduce computational costs and improve real-time applicability.

Furthermore, the study's analysis underscores the importance of factors such as booking lead time in predicting customer cancellation behavior. For instance, bookings made more than a year in advance had a cancellation rate as high as 57.14%, while bookings within a

week had a much lower rate of 7.73%. These results highlight the critical role of booking time in predicting cancellations, consistent with the findings from the CatBoost model. The study also identified important influences of customer nationality, special requests, and lead time on cancellation probabilities. Our dynamic temporal reinforcement learning model effectively captured these trends, enhancing both predictive adaptability and accuracy. The use of mutual information analysis and cross-validation techniques helped ensure the robustness of the findings.

Moreover, we introduced policy-enhanced reinforcement learning optimization techniques to further improve the effectiveness of the dynamic temporal reinforcement learning models. This allowed for better optimization of long-term returns or reductions in cancellation rates, providing hotel managers with more accurate predictions of customer behavior and helping to improve resource allocation and customer relationship management. The combination of dynamic reinforcement learning and policy optimization represents a novel approach to hotel booking predictions, offering a foundation for future research in similar applications.

## SUMMARY

This study presents an innovative approach within the framework of LSTM, aimed at addressing complex data relationships and model optimization challenges. The proposed model demonstrates enhanced predictive capability and stability through dynamic adjustments and multimodal data fusion. Experimental results showed that the model performs well in nonlinear and dynamic environments, particularly in handling long sequence data while maintaining performance stability. Additionally, we explored the application of deep learning in dynamic temporal reinforcement learning and policy-enhanced reinforcement learning optimization techniques, providing new insights and solutions for addressing practical challenges. However, the generalizability of the model should be further validated with additional datasets and scenarios. Future research could extend the applicability of these models to broader prediction tasks, enhancing both model generalization and real-world practicality.

## APPENDIX: MATHEMATICAL THEOREMS, INFERENCE PROOFS

**Theorem 3** *Assuming our Dynamic Temporal Reinforcement Learning Prediction model satisfies the update rule for state-action value function:*

$$Q(s, a) = \mathbb{E}_{\pi_\theta} \left[ r + \gamma \max_{a'} Q(s', a') \middle| s, a \right] \tag{20}$$

*This equation represents the Bellman update, a recursive formula that optimizes decision-making by considering both immediate and future rewards. The Bellman Equation is central to reinforcement learning because it allows us to decompose complex decision processes into smaller, tractable problems by recursively computing the optimal value function.*

**Proof** *Firstly, according to the Bellman equation in dynamic temporal reinforcement learning prediction, the update rule for the state-action value function $Q(s,a)$ is as follows:*

$$Q(s,a) = \mathbb{E}_{\pi_\theta}\left[r + \gamma \max_{a'} Q(s',a') \middle| s,a\right] \tag{21}$$

*Here, $r$ denotes the immediate reward under current state $s$ and action $a$, $\gamma$ is the discount factor, $s'$ is the next state, and $a'$ is the action chosen under state $s'$. This equation updates the value of a state-action pair by considering both the immediate reward and the expected reward from following the optimal policy in the future.*

*We utilize a deep neural network (MLP) to approximate the state-action value function $Q(s,a;\theta)$, updating the neural network parameters $\theta$ by minimizing the mean square error to approximate the true state-action value function. The policy gradient update rule is derived to optimize the decision policy $\pi_\theta$:*

$$\nabla_\theta J(\theta) = \mathbb{E}_{\pi_\theta}\left[\nabla_\theta \log \pi_\theta(a|s)\mathbb{E}_{\pi_\theta}\left[r + \gamma \max_{a'} Q(s',a') \middle| s,a\right]\right] \tag{22}$$

*Here, $J(\theta)$ is the objective function representing the expected cumulative reward, and $\pi_\theta$ is the decision policy parameterized by $\theta$. The gradient $\nabla_\theta J(\theta)$ is used to update the policy parameters to maximize the expected reward. Finally, the update rule for the state value function $V(s)$ is:*

$$V(s) = \mathbb{E}_{\pi_\theta}\left[r + \gamma V(s') \middle| s\right] \tag{23}$$

*This recursive formulation dynamically adjusts the value of a state by accounting for both current and future rewards. Through this process, the model can effectively predict and optimize hotel booking cancellation probabilities in dynamic environments.*

**Lemma 3** *In the framework of dynamic temporal reinforcement learning prediction technology, we define the update rule for the state value function $V(s)$ as follows:*

$$V(s) = \mathbb{E}_{\pi_\theta}\left[r + \gamma \max_{a'} \mathbb{E}_{\pi_\theta}\left[r + \gamma \max_{a''} \mathbb{E}_{\pi_\theta}\left[r + \gamma \max_{a'''} \mathbb{E}_{\pi_\theta}\left[r + \gamma \max_{a''''} V(s^{(4)}) \middle| s^{(3)}, a''' \right] \middle| s^{(2)}, a'' \right] \middle| s^{(1)}, a' \right] \tag{24}$$

*This is an expanded form of the Bellman Equation, representing the recursive update of the state value function $V(s)$. It captures how future rewards, discounted by $\gamma$, are recursively propagated back to the current state, allowing the model to predict long-term rewards.*

**Proof** *Firstly, according to the Bellman equation, we have:*

$$Q(s,a) = \mathbb{E}_{\pi_\theta}\left[r + \gamma \max_{a'} Q(s',a') \middle| s,a\right] \tag{25}$$

*This equation is the foundation of reinforcement learning, as it balances immediate rewards and future returns. The expanded form:*

$$Q(s,a) = \mathbb{E}_{\pi_\theta}\left[r + \gamma \max_{a'} \mathbb{E}_{\pi_\theta}\left[r + \gamma \max_{a''} \mathbb{E}_{\pi_\theta}\left[r + \gamma \max_{a'''} Q(s^{(3)}, a''') \middle| s^{(2)}, a'' \right] \middle| s^{(1)}, a' \right] \tag{26}$$

*captures a multi-layer recursive form, which is used to handle complex, dynamic environments by propagating value updates across multiple decision steps. This allows the model to learn optimal actions over long time horizons.*

*In the training process of dynamic temporal reinforcement learning, we use a multi-layer perceptron (MLP) to approximate the state value function $V(s)$. The approximation helps to generalize the learning process over large and complex state spaces.*

$$V(s) = \mathbb{E}_{\pi_\theta}\left[ r + \gamma \max_{a'} \mathbb{E}_{\pi_\theta}\left[ r + \gamma \max_{a''} \mathbb{E}_{\pi_\theta}\left[ r + \gamma \max_{a'''} \mathbb{E}_{\pi_\theta}\left[ r + \gamma \max_{a''''} V(s^{(4)}) \middle| s^{(3)}, a''' \right] \middle| s^{(2)}, a'' \right] \middle| s^{(1)}, a' \right] \right] \tag{27}$$

*This recursive approach allows the model to predict and optimize long-term outcomes, adapting to complex dynamics and improving decision-making strategies in real-world tasks like hotel booking cancellations.*

### Funding

The authors received no funding for this work.

### Competing Interests

The authors declare there are no competing interests.

### Author Contributions

- Junhua Xiao conceived and designed the experiments, performed the experiments, analyzed the data, performed the computation work, prepared figures and/or tables, authored or reviewed drafts of the article, and approved the final draft.
- Shahriman Zainal Abidin conceived and designed the experiments, authored or reviewed drafts of the article, and approved the final draft.
- Verly Veto Vermol performed the experiments, analyzed the data, performed the computation work, prepared figures and/or tables, authored or reviewed drafts of the article, and approved the final draft.
- Bei Gong conceived and designed the experiments, performed the experiments, prepared figures and/or tables, and approved the final draft.

### Data Availability

The third-party dataset is available at Alibaba Cloud Tianchi: https://tianchi.aliyun.com/dataset/90442.

### Supplemental Information

Supplemental information for this article can be found online at http://dx.doi.org/10.7717/peerj-cs.2442#supplemental-information.

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
