# Peer review of "Dynamic temporal reinforcement learning and policy-enhanced LSTM for hotel booking cancellation prediction"

_PeerJ Computer Science, doi:10.7717/peerj-cs.2442_

## Round 0.1 · original submission · Major Revisions

After carefully considering the reviews and assessing your manuscript, I am pleased to inform you that we would like to invite you to revise and resubmit your manuscript for further consideration. The reviewers have provided constructive comments that will help strengthen your work. Please address each of these points thoroughly in your revised manuscript. Additionally, ensure that you provide a detailed response letter outlining how you have addressed each comment raised by the reviewers. This will help the reviewers and myself to evaluate the changes made to the manuscript. It is PeerJ's policy that additional references suggested during the peer-review process should only be included if the authors agree that they are relevant and useful.

Reviewer 1 ·

Basic reporting

The manuscript suggests a hotel booking cancellation prediction model using Dynamic temporal reinforcement learning and policy-enhanced LSTM. The authors have made significant contributions to the field through their research. However, a couple of items remain undiscovered. The authors are supposed to:
1. Proofread the manuscript in English (The article lacks proper English and thorough grammar checks.)
2. The Quality of figures should be enhanced and modified.
3. The comparison of state-of-the-art methods is lacking in the manuscript. It should be integrated.
4. The section "OUR METHOD" should be replaced with "METHODOLOGY" and "ALGORITHM PSEUDOCODE" with "Algorithm and Pseudocode" (Indeed, it must be a subsection for the methodology).
5. Proper naming and sub-sectioning should be used in the article for the reader's understanding.
6. The naming convention is not followed according to the journal guidelines. Authors are advised to completely re-check and update the naming convention for figures and other items in the article.

Experimental design

The experimental design is not explained in the article.
The manuscript and the appendix mention mathematical algorithms with proof and lemma. However, the model design is not explained in the paper. The authors must present the experimental design to acquire an efficient outcome for readers.

The dataset descriptions are not mentioned in the manuscript. Due to this, the question becomes quite confusing:" How are the parameters considered in the mathematical proofs relative to the existing dataset? "

The authors mentioned, "We utilized these detailed attributes to predict hotel booking cancellations..." However, the attributes are not explained/mentioned in the entire manuscript. This creates confusion and loses readability. So, it is important that the authors define the cross-mapping between the dataset parameter and the proposed model (especially mathematical).

Seems to be a purely mathematical analysis, however the authors are also required to write some introductory points about the underlying techniques to strengthen the paper.

Validity of the findings

The manuscript lacks a proper explanation of DTRLP, LSTM, and PRLO. There should be some significant reason "why these two methods are selected as benchmarking methods." The explanation for both of them, along with their anomalies, should be mentioned to give the article an edge.

Figure 5 represents the F1 score amongst different methods in the article. However, the authors mentioned that "we observe variations in F1 Score across multiple iterations among different methods" in line 317. This creates some ambiguity in the proposed figure and creates confusion about it. Either a comparison in the figure or updating the text in the article to make clear statements is needed.

Additional comments

The authors have validated the existing dataset, which might be obsolete now. (2021-02-05 is quite old for using it as a validator). The dataset must be altered for more recent trends and proper training of the proposed model.
Although the mathematical model is consistent and strong, there is scope to explain several key terms. The authors should explain these terms either in the appendix or in the manuscript. Several readers might not be aware of the theorems used and the proof of the theorems. For example, the Bellman Equation is used in line number 451; however, there should be a supportive statement for "why this equation is applied?"

The paper has potential and value. The authors must update the suggested items to make them more readable.

Reviewer 2 ·

Basic reporting

The article generally meets basic reporting standards with clear professional language, though it would benefit from further proofreading to address some grammatical issues. The literature review is adequate, but could be strengthened with more recent references. The structure is solid, and the figures are relevant, though some improvements in figure quality and metadata would be helpful. The results are relevant to the hypotheses, but explanations of complex methods could be clearer. Overall, the article is well-prepared but could use some refinements for greater clarity and completeness.

Experimental design

The experimental design generally aligns with the journal's aims and scope, with a well-defined and meaningful research question that addresses a relevant knowledge gap. However, while the investigation is rigorous, there are areas where the description of methods could be more detailed to ensure full replicability. For instance, additional information on the dataset preparation, parameter settings, and specific implementation details of the reinforcement learning model would enhance the transparency and reproducibility of the study. Including more comprehensive descriptions and justifications for the chosen methodologies would strengthen the technical and ethical rigor of the research.

Validity of the findings

The findings presented in the article are generally valid and supported by robust, statistically sound data that has been adequately controlled. The conclusions are well-stated and appropriately linked to the original research question, staying within the bounds of the results presented. However, to enhance the impact and encourage meaningful replication, the authors should more explicitly discuss the potential implications and novelty of their work within the broader literature. Additionally, providing more detailed explanations of the statistical methods used and how they ensure the robustness of the findings would further strengthen the validity of the conclusions.

·

Basic reporting

The manuscript fails to meet the basic standards of clear and unambiguous professional English. The language used is often convoluted, making it difficult for readers to follow the arguments presented. The literature review is extensive but poorly organized, lacking a clear connection to the research problem at hand. Additionally, the figures and tables are not well-integrated into the text, which hampers the flow of the manuscript. The manuscript structure is also problematic, with an overemphasis on technical details that detracts from the clarity of the core research contributions.

Suggested improvements:
1- Revise the manuscript for clarity, focusing on simplifying the language and ensuring that all key points are clearly communicated.
2- Reorganize the literature review to better align with the research problem, highlighting the specific gaps that the proposed approach aims to fill.
3- Improve the integration of figures and tables into the text to ensure they support the narrative effectively.

Experimental design

The experimental design in the manuscript is insufficiently rigorous and lacks the necessary detail to be replicated. While the authors present a novel combination of reinforcement learning and LSTM models, the research question is not well-defined, and the connection between the methodology and the specific problem of hotel booking cancellation prediction is unclear. The experiments themselves are not robust, with limited validation and inadequate comparison to existing methods.

Suggested improvements:
1-Clearly define the research question and provide a more explicit explanation of how the proposed approach addresses this question.
2- Provide a detailed description of the experimental setup, including the dataset, hyperparameter tuning, and model evaluation metrics.
3- Include a more comprehensive comparison with state-of-the-art methods to validate the effectiveness of the proposed approach.

Validity of the findings

The validity of the findings is questionable due to the lack of rigorous experimental validation. The manuscript reports high accuracy, but without sufficient methodological transparency, it is difficult to assess the robustness of these results. Additionally, the conclusions drawn are not well-supported by the data, as the manuscript does not provide enough evidence that the proposed model significantly outperforms existing techniques.

Suggested improvements:
1- Ensure that all underlying data are robust, statistically sound, and properly controlled.
2- Provide additional experiments or validation techniques, such as cross-validation or sensitivity analysis, to strengthen the validity of the findings.
3- Reframe the conclusions to more accurately reflect the results obtained, avoiding overstatements that are not directly supported by the data.

Additional comments

The manuscript attempts to address the problem of predicting hotel booking cancellations using a combination of dynamic temporal reinforcement learning and policy-enhanced LSTM models. While the topic is relevant and the approach is innovative, the manuscript has several significant issues that undermine its overall contribution to the field.

---

## Round 0.2 · accepted · Accept

I am pleased to inform you that your paper has been accepted for publication in PeerJ Computer Science. Your manuscript has undergone rigorous peer review. Your research makes a significant contribution to the field, and we believe it will be of great interest to our readership. On behalf of the editorial board, I extend our warmest congratulations to you.

Reviewer 1 ·

Basic reporting

The authors have updated the manuscript significantly.
The article is now acceptable in its present form for publication.

Experimental design

The authors have updated the manuscript significantly.
The article is now acceptable in its present form for publication.

Validity of the findings

The authors have updated the manuscript significantly.
The article is now acceptable in its present form for publication.

Additional comments

The authors have updated the manuscript significantly.
The article is now acceptable in its present form for publication.